depression; anxiety; peer support; refugee; global mental health

**Corresponding author:**
Michalis Lavdas;
Email: michail.lavdas@uib.no

‡This article has been updated since original publication. A notice detailing the change has been published.

# Mental health trajectories and Peer Refugee Helper engagement, among Afghan, Iranian and Syrian refugees and asylum seekers in Greece‡

Michalis Lavdas[1] , Gro Mjeldheim Sandal[1], Marit Sijbrandij[2] ,
Trynke Hoekstra[3] and Tormod Bøe[1]

[1]Department of Psychosocial Science, Faculty of Psychology, University of Bergen, Bergen, Norway; [2]Department of Clinical, Neuro- and Developmental Psychology, WHO Collaborating Center for Research and Dissemination of Psychological Interventions, Amsterdam Public Health research institute, Vrije Universiteit Amsterdam, Amsterdam, The Netherlands and [3]Department of Health Sciences and Amsterdam Public Health research institute, Vrije Universiteit Amsterdam, Amsterdam, The Netherlands

## Abstract

Peer Refugee Helpers (PRHs) support peers in humanitarian settings, which may influence their own mental health. This longitudinal study examined anxiety and depression trajectories among Afghan, Iranian and Syrian refugees and asylum seekers in Greece, focusing on how PRH status (paid/unpaid) and sense of coherence influence trajectory membership. The study included 176 adult, PRHs and non-helpers. The following scales were administered three times at ~4-month intervals: Patient Health Questionnaire (PHQ-9), Generalized Anxiety Disorder (GAD-7), Social Provisions Scale (SPS-24), Sense of Coherence (SOC-13), Perceived Ability to Cope With Trauma (PACT) and Brief Trauma Questionnaire (BTQ). Using latent growth mixture modeling, we identified two depression (high and low) and three anxiety (high, moderate and low) trajectories. The adjusted logistic and multinomial regression models indicated that unpaid PRHs were significantly less likely to follow a low depression trajectory (odds ratio [OR] = 0.55, $p = 0.037$), while paid PRHs were more likely to follow a low anxiety trajectory (OR = 3.17, $p = 0.009$). Higher SOC was associated with low depression (OR = 1.03, $p = 0.012$) and low anxiety trajectories (OR = 1.06, $p = 0.002$). Our findings suggest PRH mental health may be associated with working conditions, including financial compensation.

## Impact statement

This longitudinal study examined the mental health trajectories of Peer Refugee Helpers (PRHs), that is, refugees and asylum seekers supporting their peers in humanitarian settings. Despite their important role in delivering task-shared mental health and psychosocial support (MHPSS) interventions, PRHs remain an understudied group of aid workers. Operating under precarious conditions, they face substantial mental health challenges.

To our knowledge, this is the first longitudinal study of PRHs, where we identified unique stable trajectories in anxiety and depression influenced by payment status and sense of coherence (SOC). Our findings show that unpaid PRHs were significantly more likely to follow a high depression trajectory (i.e., scoring high on a measure of depression symptoms), while paid PRHs were more likely to follow a low anxiety trajectory (i.e., scoring low on a measure of anxiety symptoms). Higher SOC was further associated with low depression and low anxiety trajectories. Additionally, being a woman and past trauma exposure reduced the likelihood of a low depression and low anxiety trajectory, whereas social support increased it.

These results carry important implications for policy and practice. First, SOC and PRH mental health may be strengthened through integration of task-sharing interventions, particularly when combined with training and supervision. Second, financial compensation and increased employability opportunities may further protect PRH's mental health. Finally, MHPSS interventions should be particularly attentive to mental health among women involved as PRHs, ensuring their safety and well-being.

## Introduction

Forcibly displaced people in the world, including refugees and asylum seekers, reached 117.3 million at the end of 2023 (United Nations High Commissioner for Refugees, 2024). Refugees and asylum seekers experience a high prevalence of mental health problems and barriers that might

hinder access to mental health and psychosocial support (MHPSS) (Fuhr et al., 2019; Kiselev et al., 2020; Posselt et al., 2020).

Peer Refugee Helpers (PRHs) are forcibly displaced individuals formally affiliated with an Aid/Humanitarian Organization (AO) in paid or unpaid roles (de Graaff et al., 2020; Lavdas et al., 2024a). PRHs have a central role in the humanitarian field, providing different types of psychological support to their peers, including scalable interventions such as Problem Management Plus (PM+) (Spaaij et al., 2023; Surkan et al., 2024). PRHs may further work as interpreters (Kindermann et al., 2017), community health workers among specific refugee groups (Izugbara et al., 2018) and peer mentors (Paloma et al., 2020).

We conducted the current study in Greece, as it is one of the major entry points to Europe (United Nations High Commissioner for Refugees, 2025) and thus highly relevant to the broader humanitarian context. PRHs in Greece reported moderately high levels of depression and mild to moderate levels of anxiety, similar to non-helpers in a cross-sectional study of Lavdas et al. (2024a), which used baseline data from the current longitudinal study. Serving as a PRH may have both salutary and adverse effects on the individual, depending on the personal and environmental resources available (Lavdas et al., 2024b). While non-specialist providers of task-sharing interventions, such as PRHs, may promote their own mental health and well-being by applying the strategies they learn, they may also experience harmful effects associated with challenges such as gender-based violence and low pay (Sangraula et al., 2024).

Past research suggests that taking up a PRH role might benefit mental health. A qualitative study including PRHs with Syrian refugee backgrounds in Switzerland found that the PRH role could lead to more resilient responses (Spaaij et al., 2023). Specifically, the PRHs reported less negative impact from clients' stressful experiences than before engaging as PRHs. PRHs further reported that self-management strategies learned through the PM+ intervention benefited their own mental health (Spaaij et al., 2023).

An important framework for understanding how individuals protect their health is sense of coherence (SOC), a fundamental concept in the salutogenic model of health (Antonovsky, 1979). SOC consists of three components: comprehensibility (perceiving challenges as understandable), manageability (the level of control one perceives they have over such challenges) and meaningfulness (experiencing these challenges as worthy of engagement and mobilizing appropriate resources) (Antonovsky, 1993). SOC has been identified as an important resource that is influenced through a combination of personal, social and environmental factors, including the work environment and helps protect against adverse work-related outcomes, such as stress and burnout (Jenny et al., 2017). Being a PRH appears to increase SOC through feeling empowered, learning new skills and experiencing role clarity and safety (Lavdas et al., 2024b). Increased SOC and financial compensation were associated with significantly lower levels of anxiety and depression among PRHs in the study of Lavdas et al. (2024a). SOC has been found to protect mental health among non-specialist local aid workers (Veronese and Pepe, 2017) and to positively contribute to post-assignment health among international aid workers (de Jong et al., 2022). Using growth mixture modeling, de Jong et al. (2023) identified latent trajectories among international aid workers, reporting that SOC, coping self-efficacy and social support were associated with better mental health, while being a woman and length of assignment contributed negatively. The PRH role has been associated with a negative impact on mental health and well-being as well. For example, PM+ helpers with migrant backgrounds in France reported emotional overload and frustration during

service delivery in cases when they were unable to distance themselves from their clients (Surkan et al., 2024). Similarly, secondary traumatization among PRHs was raised as an important issue during implementation in the Netherlands (Woodward et al., 2022). PRHs in Greece identified job insecurity and low, inconsistent payment as significant hardships that made it difficult to establish boundaries in their interactions with their peers (Lavdas et al., 2024b). PRHs delivering a scalable intervention for adolescents in Lebanon additionally reported frustration due to a lack of job security, unstable hours and limited opportunities for career development (Ali et al., 2024).

Lavdas et al. (2024a) found a positive association between the number of potentially traumatic events experienced and depression among PRHs, which attenuated in the fully adjusted model. Having experienced a higher number of potentially traumatic events predicted increased symptom severity in depression among aid workers in Germany, professional and volunteer, of local and migrant backgrounds (Borho et al., 2019). Among PRHs, past trauma exposure may be associated with adverse reactions during service provision and increased role overload (Lavdas et al., 2024b). Including past trauma exposure in the analysis is important as it helps determine whether the psychological impact of being a PRH is related to personal experiences of potentially traumatic events or may be solely attributed to the PRH role or work environment.

There is a gendered dimension in humanitarian work, shaped by structural inequalities, according to Strohmeier and Panter-Brick (2022). According to Crenshaw (1991, 1245), the effects of discrimination are best understood when taking into consideration the "multiple grounds of identity," which include gender, race, class and sexuality. Within the group of international aid workers that often live disconnected from the local communities, women report difficulties balancing family and work obligations, which may hinder career progression (Strohmeier and Panter-Brick, 2022). In another study among international aid workers, Martinmäki et al. (2023) found significantly higher rates of experienced workplace sexual harassment among women, which were associated with anxiety and depression. Female non-specialist providers with local backgrounds further reported experiencing sexual harassment and a lack of safety at work in the review of Sangraula et al. (2024). Several forms of gender-based violence (GBV), such as physical abuse of the wife by her husband, may be invisible or normalized within communities, which could further present unique challenges in reporting such incidents and addressing them within the humanitarian field (Izugbara et al., 2018). Female Afghan PRHs in urban and camp settings in Greece reported experiencing role conflict as a result of socio-cultural gender norms that women were expected to abide by and GBV experienced in their workplace (Lavdas et al., 2024b). Taken together, the abovementioned findings indicate that the challenges that female PRHs may experience may be shaped by intersecting experiences and identities.

## Aim of the study

There are both qualitative (Woodward et al., 2022; Spaaij et al., 2023; Lavdas et al., 2024b) and quantitative studies (Lavdas et al., 2024a) on PRH mental health. However, to our knowledge, no longitudinal studies have examined mental health trajectories of PRHs. This study addresses this gap, aiming first to identify anxiety and depression trajectories among Afghan, Iranian and Syrian refugees and asylum seekers in Greece using latent growth mixture modeling (LGMM), and second, to examine how PRH status and SOC influence trajectory membership. In this study, we further

adjust for important covariates identified in the literature, including age, gender, education, years in Greece, exposure to traumatic events, coping flexibility and social support.

We hypothesized that (1) being a paid PRH and (2) having a higher SOC would be associated with fewer anxiety and depression symptoms.

## Methods

### Participants

Participants were recruited through AOs, which operated community centers in Greece and provided services for refugees and asylum seekers, such as language courses and MHPSS. The participants resided in urban areas or refugee camps in Attica or Northern Greece and included 176 refugees or asylum seekers (18+ years) from Afghanistan, Syria or Iran, with or without PRH status, who took part in at least two of three data collection phases. Participants were required to understand Dari, Farsi or Arabic. PRH status was defined as formal affiliation with an AO with or without financial compensation and could vary over data collection (see Table 1).

We estimated the required sample size using G*Power version 3.1.9.2, setting alpha to 0.05 (two-tailed) and power (1-cc) to

**Table 1.** Descriptive statistics at baseline

| Factors | N | N = 153[1] |
|---|---|---|
| **Age** | 153 | 34.9 (9.1) |
| **Gender** | 153 | |
| Male | | 68 (44.4%) |
| Female | | 80 (52.3%) |
| Non-binary/not disclosed | | 5 (3.3%) |
| **Country of origin** | 151 | |
| Afghanistan | | 39 (25.8%) |
| Iran | | 55 (36.4%) |
| Other | | 24 (15.9%) |
| Syrian Arab Republic | | 33 (22.9%) |
| Missing | | 2 |
| **Years in school** | 152 | 10.7 (5.8) |
| Missing | | 1 |
| **Years in Greece** | 151 | 7.2 (6.3) |
| Missing | | 2 |
| **Children** | 125 | |
| No children | | 32 (26.6%) |
| 1–2 children | | 63 (50.4%) |
| 3–4 children | | 23 (18.4%) |
| More than 4 | | 7 (5.6%) |
| Missing | | 28 |
| **Legal status** | 152 | |
| Refugee status | | 88 (57.9%) |
| Asylum seeker | | 45 (29.6%) |

*(Continued)*

**Table 1.** (*Continued*)

| Factors | N | N = 153[1] |
|---|---|---|
| Other | | 19 (12.5%) |
| Missing | | 1 |
| **Occupation** | 148 | |
| Other | | 42 (28.4%) |
| Paid work | | 39 (26.4%) |
| Unemployed | | 52 (35.1%) |
| Volunteer | | 15 (10.1%) |
| Missing | | 5 |
| **Number of traumatic events** | 153 | 3.4 (2.5) |
| **PRH status** | 148 | |
| Paid PRH | | 29 (19.6%) |
| Paid non-helper | | 10 (6.8%) |
| Unpaid PRH | | 48 (32.4%) |
| Unpaid non-helper | | 61 (41.2%) |
| Missing | | 5 |
| **Living situation** | 152 | |
| Living without a partner | | 78 (51.3%) |
| Living with a partner | | 74 (48.7%) |
| Missing | | 1 |
| **Accommodation** | 113 | |
| Refugee camp | | 14 (12.4%) |
| Urban area | | 83 (73.5%) |
| Other | | 16 (14.2%) |
| Missing | | 40 |
| **GAD–7 score** | 150 | 8.87 (5.74) |
| Missing | | 3 |
| **PHQ–9 score** | 150 | 11.07 (6.97) |
| Missing | | 3 |
| **SOC score** | 149 | 49.6 (12.9) |
| Missing | | 4 |
| **SPS score** | 146 | 65.9 (12.8) |
| Missing | | 7 |
| **PACT score** | 146 | 0.6 (0.6) |
| Missing | | 7 |

[1]Mean (SD); *n* (%).
*Note.* GAD-7, generalized anxiety disorder scale; PACT, Perceived Ability to Cope With Trauma; PHQ-9, Patient Health Questionnaire; PRH, Peer Refugee Helper; SOC-13, sense of coherence; SPS, Social Provisions Scale.

80. To detect a small effect size of Cohen's $f = 0.2$, 100 participants were needed.

### Procedure

This longitudinal study included three data collection phases at ~4-month intervals. Baseline (T1) occurred from November 24, 2022, to January 29, 2023; T2 from May 24 to June 26, 2023; and T3 from December 15, 2023, to February 8, 2024. Surveys were

created and administered using Qualtrics (https://www.qualtrics.com), with an average completion time of 28 min, on mobile phones and computers connected to the internet. The survey was available in Dari/Farsi, Arabic and English.

At T1, 248 participants enrolled. Seven respondents required face-to-face assistance due to illiteracy or disability, which was provided by a research assistant. Data collected at T1 were also used by Lavdas et al. (2024a). At T2, 167 participants enrolled (126 returning from T1 and 41 new entrants), among whom 7 required in-person assistance. Dropout reasons included moving out of Greece and being no longer interested in participating, communicated directly as a response to follow-up invitation, no valid contact information or no answer without further justification. At T3, 151 participants enrolled (105 from T1, 23 from T2 and 23 new). All T3 surveys were completed online. In total, 78 participants participated in all three time points, and 176 participated in at least two time points.

We involved two native Arabic and three native Dari- and Farsi-speaking research assistants with lived experience in forced displacement in adapting the instruments and in data collection. Research assistants were reimbursed for their contribution. The first author, a clinical psychologist and MHPSS specialist, met regularly with the research assistants, online or face-to-face throughout the study, to discuss emerging issues and provide a safe space for emotional support. In cases of participants requesting further assistance, the first author and the research assistants discussed appropriate solutions and provided relevant contacts for resources and services. We further established a reference group consisting of stakeholders in the humanitarian field, such as representatives from non-governmental organizations, refugee communities, academia and international organizations. The reference group facilitated recruitment and dissemination of the study and participated in discussions for the interpretation of the findings and policy implications.

### Measures

#### PRH status and demographics

We used a demographic questionnaire that included one question on occupational status and a separate question on PRH experience: "Have you had any experience helping other refugees in a formal position at an organization or service?" Those who answered positively ("yes") received further questions on payment status (paid/unpaid), their tasks and roles. Combining occupational and PRH status, we created four categories (paid/unpaid PRHs and paid/unpaid non-helpers; see Table 1). The questionnaire further included questions on demographics such as age, gender, education, country of origin, years in school and years lived in Greece.

#### Trauma exposure

The Brief Trauma Questionnaire (BTQ) (Schnurr et al., 1999) measures exposure to various potentially traumatic events. The 10 items examine experiencing war, serious accidents, natural or technological disasters, life-threatening illnesses, physical abuse, attacks, sexual abuse, serious injury or fear of being seriously injured or killed, loss of a family member or friend and witnessing or fearing for someone being seriously injured or killed. To quantify the number of exposures to potentially traumatic events, we calculated the sum of affirmative answers to the question "Has this ever happened to you?" (yes/no) for each item (score range: 1–10). The BTQ was available in English and was translated into Arabic and Dari for our study.

#### Mental health outcomes

We assessed anxiety using the Generalized Anxiety Disorder Scale (GAD-7) (Spitzer et al., 2006). Example items include "Not being able to stop or control worrying" and "Becoming easily annoyed or irritable." Frequency was assessed using a Likert scale (0 = *Not at all*, 1 = *Several days*, 2 = *More than half the days* and 3 = *Nearly every day*). The GAD-7 includes a total of seven items assessing anxiety symptoms, with the total score being the sum of the scores in each item. Higher total scores indicate greater anxiety, categorized as minimal (0–4), mild (5–9), moderate (10–14) or severe (15+). Cutoff scores informed trajectory interpretation. We further used the Arabic (Sawaya et al., 2016) and Dari (Perera and Lavdas, 2020) GAD-7 versions. In the current study, the GAD-7 showed excellent internal consistency, with Cronbach's alpha values of 0.89 at T1, 0.93 at T2 and 0.91 at T3. Omega analysis revealed strong reliability, with Omega Total values ranging from 0.92 at T1, 0.94 at T2 and 0.94 at T3 and Omega Hierarchical values consistently above 0.79 across all time points.

We used the Patient Health Questionnaire (PHQ-9) (Kroenke et al., 2001) to assess depression. Example items include "Feeling tired or having little energy" and "Thoughts that you would be better off dead or of hurting yourself in some way." A Likert scale is used to assess frequency (0 = *Not at all*, 1 = *Several days*, 2 = *More than half the days* and 3 = *Nearly every day*). The PHQ-9 has a total of nine items, and the total score is the sum of all items. Higher scores indicate higher depression, categorized as minimal (0–4), mild (5–9), moderate (10–14), moderately severe (15–19) and severe (20–27). The cutoff scores informed the interpretation of the trajectories. We used the Arabic version from Sawaya et al. (2016) and the Farsi (Dadfar et al., 2018) version of PHQ-9. The PHQ-9 demonstrated excellent internal consistency in this study, with Cronbach's alpha values of 0.89 at T1, 0.91 at T2 and 0.92 at T3. Omega analysis also revealed strong reliability, with Omega Total values ranging from 0.92 at T1, 0.93 at T2 and 0.94 at T3 and Omega Hierarchical values consistently above 0.77 across all time points.

#### Social support and resources

Social support was assessed through the Social Provision Scale (SPS) (Cutrona and Russell, 1987). The SPS-24 includes 24 items, and higher scores indicate better social provisions. Example items include "There are people who like the same social activities I do" and "I have a trustworthy person to turn to if I have problems." A Likert scale from 1 (*strongly disagree*) to 4 (*strongly agree*) is used to assess social provisions. SPS-24 showed high internal consistency throughout the various time points, with $\alpha$ = 0.894 (T1), 0.873 (T2) and 0.861 (T3), and $\omega$ = 0.886 (T1), 0.844 (T2) and 0.819 (T3). SPS-24 was translated into Arabic and Dari for the purposes of our study.

#### Sense of coherence

We measured SOC through the SOC-13 scale (Antonovsky, 1993) containing the three components: (a) meaningfulness, (b) manageability and (c) comprehensibility. A total of 13 items is included, for example: "How often do you have the feeling that there's little meaning in the things you do in your daily life?" and "Do you have the feeling that you're being treated unfairly?" A Likert scale from 1 to 7 is used to assess responses to the items in SOC. Five items are reversed before computing the sum, with higher scores indicating a higher SOC levels. Calculations of internal consistency showed an $\alpha$ of 0.77 (T1), 0.69 (T2) and 0.74 (T3), and $\omega$ of 0.84 (T1), 0.80 (T2) and 0.83 (T3). The alpha reflects good internal consistency,

close to the norms identified by Eriksson and Lindström (2005), from 0.70 to 0.95. SoC-13 was available in Arabic (Alharbi et al., 2022) and was translated into Dari as part of this study.

### Coping flexibility

We used the Perceived Ability to Cope With Trauma (PACT) scale (Bonanno et al., 2011) to measure coping flexibility. PACT consists of two dimensions: (a) perceived ability to trauma processing and (b) moving forward after dealing with a traumatic event. Example items include "Stay focused on my current goals and plans" and "Pay attention to the distressing feelings that result from the event." Items are answered on a Likert scale (1 = *not at all able* to 7 = *extremely able*), with higher scores indicating a better ability to cope. The reliability of the PACT was high throughout the measurements: T1 ($\alpha = 0.95$, $\omega = 0.95$), T2 ($\alpha = 0.95$, $\omega = 0.95$) and T3 ($\alpha = 0.95$, $\omega = 0.95$). PACT was available in English and was translated into Arabic and Dari for this study.

### Translation and cultural adaptation of the scales

All three data collection phases used the same translated and adapted scales. We followed a three-step process for translation and adaptation, informed by Perera et al. (2020) and the World Health Organization (2016): (1) professional translation from English to the target languages (Arabic and/or Dari); (2) back-translation with experienced PRHs of Syrian and/or Afghan origin; and (3) review and final language adjustments by the authors and the PRHs. In the translation and adaptation, specific attention was given to items associated with cultural stigma (e.g., BTQ-10, item 7 on "unwanted sexual contact") and the use of accurate and appropriate wording in Arabic and Dari. Additionally, working with the research assistants, we discussed at length appropriate terminology to translate the "non-binary" option for gender or the "housekeeper," which was a gendered term in the initial translation and was adapted to reflect a gender-inclusive occupation. The questionnaires were further reviewed for the use of appropriate mental health terminology. Metaphors were given specific attention (e.g., PACT, item 3, "look for a silver lining" translated into "look for the good thing/positive in the bad").

### Statistical analyses

Data cleaning, scoring and calculating measure reliability were conducted using SPSS and R (version 4.3.0; R Core Team, 2023). Reliability calculations were performed on the combined data from the translated and English versions of the scales. We followed the GRoLTS-Checklist (van de Schoot et al., 2017) for reporting latent trajectories, using the standard three-step method. First, we determined the optimal number of classes without other predictors identifying distinct trajectories. Second, we saved the most likely class membership for each individual and merged this as a new variable with the original dataset. Third, we performed logistic regression for depression trajectories (high and low) and multinomial regression for anxiety (high, moderate and low), adjusting for potential confounders (age, gender, education, years in Greece, exposure to traumatic events, coping flexibility and social support). We used *lcmm* package for the latent class analysis (Proust-Lima et al., 2015), the *rms* package (Harrell, 2023) for logistic regressions and the *nnet* package for multinomial regressions (Venables and Ripley, 2002). We further used *sjPlot* (Lüdecke, 2024) for visualizing model coefficients, and the *psych* r package (Revelle, 2024) for reliability calculations.

For scoring the PHQ-9 and GAD-7, we used mean imputation at the item level for those items that were partially completed (missing data >2%). Missing data on the outcome variables (GAD-7 and PHQ-9) followed a missing at random pattern and were handled through full information maximum likelihood in the latent trajectory analysis with the *lcmm* r package. We further employed multiple imputations ($m = 40$) with the *mice* r package (van Buuren and Groothuis-Oudshoorn, 2011) for missing data in the covariates in the multinomial regression in GAD-7 and with the *rms* r package in the logistic regressions for PHQ-9. Missing data for the covariates ranged from 1.2% to 13.1%. We treated the variables: age, gender, country of origin, years in Greece and years in school, as stable over the data collection phases and applied a coalescing method, retaining the first valid input for each participant and imputing it across the remaining data collection phases.

At T2, we added 23 individuals who also participated at T3 after comparing their baseline with the initial cohort from T1. Before we integrated late entrants in our study at T2, we first ran a preliminary analysis comparing them with the original sample. We compared the original sample to the 23 additional participants at T2 in terms of baseline characteristics to identify any significant differences that should be taken into consideration in the analysis. Those added at T2 had been staying in Greece for about a year longer (8.22), had higher scores in coping flexibility (1.09) and had a more diverse accommodation profile than the original cohort (Supplementary Material). Comparison was done through the Wilcoxon rank-sum test for years lived in Greece and PACT scores, and Fisher's exact test was used for the accommodation profile. Although the above differences were significant, none of the factors included in the analysis proved significant in the fully adjusted models.

The primary method used for analysis was LGMM, implemented with the *lcmm* r package and specifically the heterogeneous linear mixed model (*hlme*) function. Using Shapiro–Wilk normality tests, we confirmed that our continuous outcome variables (GAD-7, PHQ-9) had a non-normal distribution. Since the *hlme* function assumes normal distribution for optimal performance, we first normalized the data using the Box–Cox transformation method. We initially fitted models with only a random intercept, then introduced a random slope to capture more complexity in the data, without allowing covariances to vary. The inclusion of the random slope improved the model's explanatory power. Symptoms of mental health problems do not always develop in a linear fashion (Bonanno, 2008), and a quadratic time structure was adopted. We further performed an automatic grid search for a maximum of 20 iterations from 100 random initial value vectors, ensuring model stability.

We tested models with one to five latent classes for each outcome variable. To determine the optimal number of classes, we used fit criteria: Bayesian Information Criterion (BIC), Akaike Information Criterion (AIC), sample-size adjusted BIC and entropy. We selected the model with the highest entropy and the lowest possible BIC and AIC, ensuring meaningful class separation (i.e., with class sizes >15%) and convergence. We included entropy since, according to van de Schoot et al. (2017), the three-step method that we applied is based on the assumption that class allocation occurs without classification errors and a high-enough entropy is required. As part of a sensitivity analysis, we ran the same models on the full sample of individuals who participated at least once ($N = 312$). However, the results showed poorer entropy and less distinct trajectories for anxiety and depression. We, therefore, retained the sample with at least two timepoints, acknowledging that our

results might underestimate the prevalence of high anxiety and depression trajectories.

We merged the most likely class membership for each of the participants with the original database, separately for anxiety and depression. The trajectories were visualized using the *ggplot2* r package (Wickham, 2011). We have used ChatGPT (OpenAI, 2025) strictly for limited revision of language and to facilitate the coding process in the statistical analysis phase.

## Results

### Description of demographics

Table 1 presents baseline characteristics for individuals included in T1. The mean age was 34.9 (standard deviation [SD] = 9.1) years, while there was a balance between men (44%) and women (52%) participating in the study. Most of the participants originated from either Iran (36%) or Afghanistan (26%), with 22% coming from Syria. The average years in school was 10.7 (SD = 5.8), and the average number of years in Greece was 7.2 (SD = 6.3). Half of the participants (50%) had one to two children, and most of them had approved refugee status (58%). In terms of occupation, about one-third registered as unemployed, with one in four reporting paid employment. About 28% reported other types of occupation, such as being a student, retired, receiving benefits or a homemaker. About half of the participants reported having a partner. Most participants reported living in urban areas (73%).

### Trajectories for anxiety and depression

We found a *high* and a *low* depression trajectory for PHQ-9, and three trajectories for GAD-7, *high*, *moderate* and *low* anxiety (see Figure 1). For clarity, we first show the individual trajectories in Figure 1, which demonstrates how each individual scored in GAD-7 and PHQ-9 throughout the data collection phases. Additionally, we visualized the mean score trajectories in Figure 2. While mean scores fluctuated in both anxiety and depression scores across time points, these differences were not statistically significant, suggesting stable symptom trajectories. Trajectory labels were informed by established clinical cutoffs for both GAD-7 and PHQ-9.

### Logistic regression analysis for PHQ-9

Using the most likely class membership as an outcome variable (low, compared to high depression), PRH status and SOC were regressed on trajectory membership. The different levels of PRH status (paid or unpaid PRH and paid non-helper) were compared with the unpaid non-helper (see Table 2).

In the bivariate regression models, being a paid PRH increased the probability of belonging to the low depression trajectory (odds ratio [OR] = 2.57, 95% confidence interval [CI]: 1.47–4.48, *p* = 0.001), while being unpaid decreased this probability (OR = 0.47, 95% CI: 0.28–0.79, *p* = 0.004). Higher SOC scores were significantly associated with the low depression trajectory (OR = 1.05, 95% CI: 1.03–1.06, *p* <0.001).

In the fully adjusted model, SOC remained as a significant predictor of belonging to the low depression trajectory (OR = 1.03, 95% CI: 1.01–1.05, *p* = 0.012). Higher SPS scores similarly increased the probability for the low depression trajectory (OR = 1.03, 95% CI: 1.01–1.06, *p* = 0.005).

Unpaid PRH status remained significantly associated with increased probability for higher depression symptoms (OR = 0.55, 95% CI: 0.32–0.96, *p* = 0.037). Female gender was associated with reduced odds ratio of being in the low depression group compared to the high depression group (OR = 0.47, 95% CI: 0.29–0.75, *p* = 0.002). A higher number of traumatic events reduced the likelihood of a low depression trajectory (OR = 0.83, 95% CI: 0.75–0.92, *p* < 0.001).

### Multinomial regression analysis for GAD-7

In our bivariate models, we separately regressed PRH status (paid or unpaid PRH and paid or unpaid non-helper) and SOC on trajectory membership (high, moderate or low anxiety), with high anxiety being the reference category (see Table 3). We then controlled for several covariates in the fully adjusted models.

In the bivariate regression models, being a paid PRH and having higher SOC were associated with fewer anxiety symptoms over time. For the low versus high anxiety trajectory, being a paid PRH (OR = 3.51, 95% CI: 1.61–7.65, *p* = 0.002) and having high SOC (OR = 1.09, 95% CI: 1.06–1.12, *p* <0.001) protected against anxiety symptoms. For the moderate versus high anxiety trajectory, high SOC remained significant (OR = 1.05, 95% CI: 1.02–1.07, *p* <0.001), increasing the probability of belonging to the moderate trajectory.

In the fully adjusted models, paid PRH work remained significant for predicting a low versus high anxiety trajectory (OR = 3.17, 95% CI: 1.33–7.52, *p* = 0.009). High SOC remained significant for both low versus high (OR = 1.06, 95% CI: 1.02–1.10, *p* = 0.002), as well as moderate versus high anxiety trajectories (OR = 1.03, 95% CI: 1.00–1.06, *p* = 0.027).

Exposure to traumatic events reduced the likelihood of low versus high (OR = 0.81, 95% CI: 0.71–0.92, *p* = 0.001), and of moderate versus high anxiety trajectories (OR = 0.88, 95% CI: 0.79–0.97, *p* = 0.008). Additionally, women were less likely than men to belong to the low versus high anxiety trajectory (OR = 0.49, 95% CI: 0.26–0.92, *p* = 0.027).

## Discussion

In this longitudinal study, we aimed to identify anxiety and depression trajectories among a sample of Afghan, Iranian and Syrian refugees and asylum seekers in Greece. We identified two latent trajectories for depression and three latent trajectories for anxiety. Being a paid PRH with high SOC significantly increased the probability of following a low anxiety trajectory. High SOC further predicted belonging to the low depression trajectory. Conversely, being an unpaid PRH decreased the probability of following a low depression trajectory.

The negative association of being an unpaid PRH on the longitudinal measure of depression symptoms is consistent with our previous findings, which only analyzed baseline measurements from T1. Unpaid PRHs had significantly higher anxiety and depression scores compared to those with paid PRH roles (Lavdas et al., 2024a). In the current longitudinal study, this association remained for depression but not for anxiety.

Being a paid PRH increases the probability for a low depression trajectory by 157% for depression scores and 251% for anxiety (*p* < 0.05). This effect may be attributed to the recognition that payment signifies acknowledgment of one's work, which again may boost mental health. In other research among peer workers,

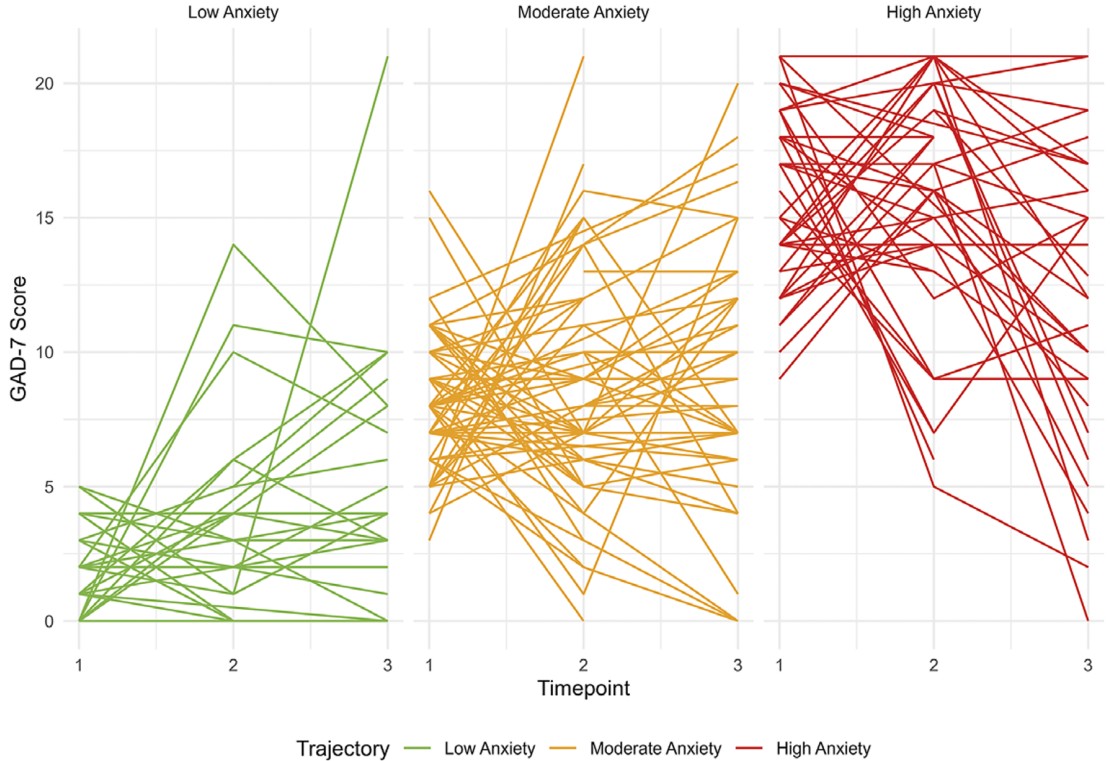

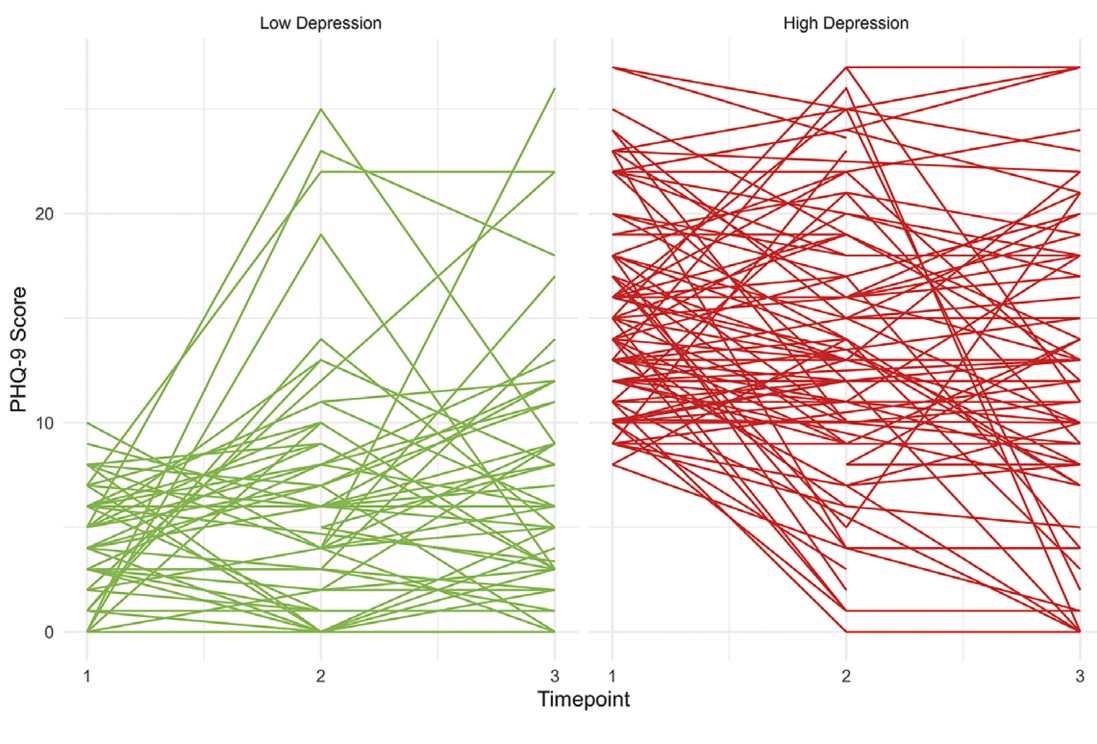

**Figure 1.** (A) The individual scores for GAD-7 (indicated by lines) as a function of the three time points in each of the three identified trajectories (presented in three separate panels in the figure). Similarly, (B) shows, in the same manner, the individual scores for PHQ-9, presented in two separate panels, corresponding to the two identified trajectories.

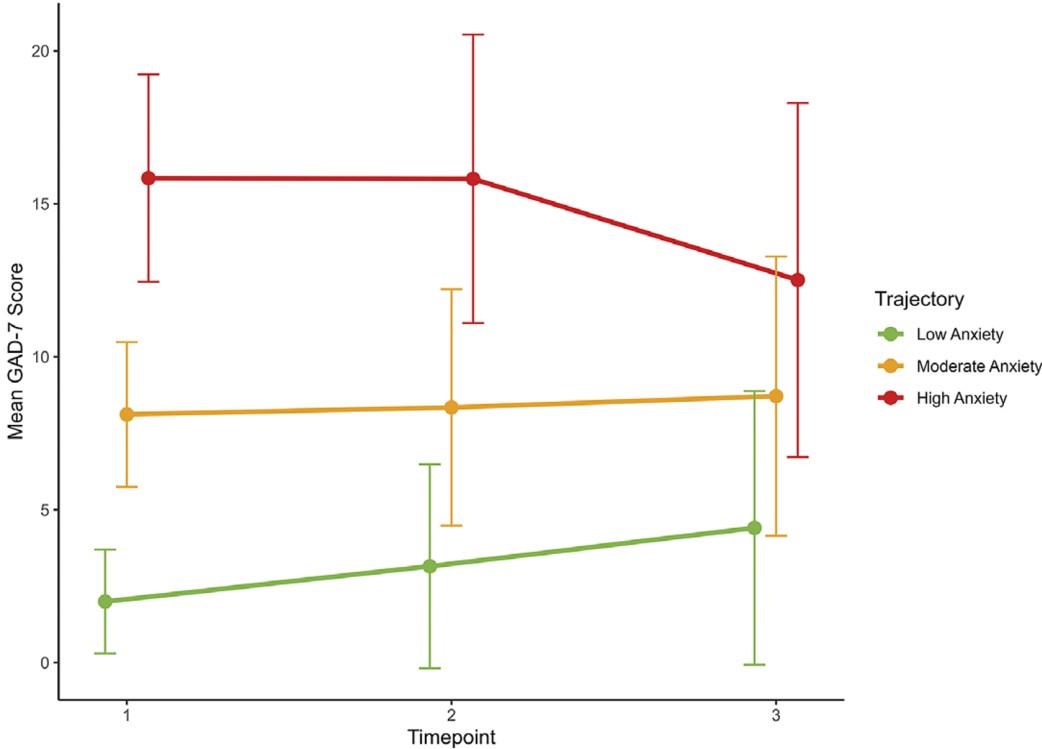

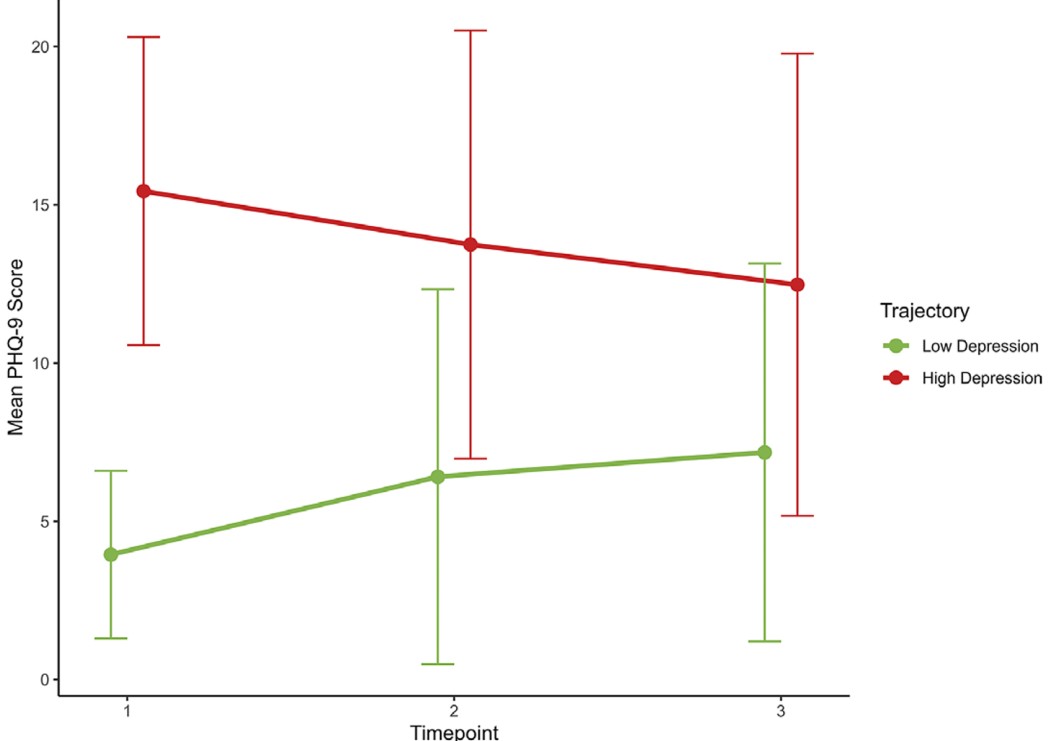

**Figure 2.** (A) The mean GAD-7 scores across three time points for each of the three identified anxiety trajectories: low, moderate and high. The *y*-axis indicates the scores on the GAD-7 scale. (B) The mean PHQ-9 scores over three time points for the two identified depression trajectories: low and high.

**Table 2.** Odds ratios for membership to the low depression trajectory for bivariate and fully adjusted models

| Characteristic | Bivariate models | | | Fully adjusted models | | |
|---|---|---|---|---|---|---|
| | OR | 95% CI | *p*-value | OR | 95% CI | *p*-value |
| **PRH status** | | | | | | |
| Intercept | 0.65 | 0.49–0.88 | **0.005** | 0.02 | 0.00–0.14 | **<0.001** |
| Unpaid non-helper | *Ref.* | *Ref.* | | *Ref.* | *Ref.* | |
| **Paid PRH** | **2.57** | **1.47–4.48** | **0.001** | 1.86 | 1.00–3.48 | 0.051 |
| Paid non-helper | 1.52 | 0.71–3.24 | 0.276 | 0.99 | 0.41–2.41 | 0.988 |
| **Unpaid PRH** | **0.47** | **0.28–0.79** | **0.004** | **0.55** | **0.32–0.96** | **0.037** |
| **SOC** | | | | | | |
| Intercept | 0.07 | 0.03–0.18 | **<0.001** | | | |
| **SOC total** | **1.05** | **1.03–1.06** | **<0.001** | **1.03** | **1.01–1.05** | **0.012** |
| Gender | | | | | | |
| Male | | | | *Ref.* | *Ref.* | |
| **Female** | | | | **0.47** | **0.29–0.75** | **0.002** |
| Non-binary/not disclosed | | | | 0.30 | 0.07–1.31 | 0.109 |
| Age | | | | 1.02 | 0.99–1.05 | 0.134 |
| Years in school | | | | 1.02 | 0.98–1.06 | 0.298 |
| **Number of traumatic events** | | | | **0.83** | **0.75–0.92** | **<0.001** |
| **SPS total** | | | | **1.03** | **1.01–1.06** | **0.005** |
| PACT | | | | 1.04 | 0.71–1.54 | 0.832 |
| Years in Greece | | | | 1.01 | 0.97–1.05 | 0.652 |

*Note.* CI, confidence interval; GAD-7, generalized anxiety disorder scale; OR, odds ratio; PACT, Perceived Ability to Cope With Trauma; PHQ-9, Patient Health Questionnaire; PRH, Peer Refugee Helper; Ref., reference; SOC-13, sense of coherence; SPS, Social Provisions Scale. Bold values indicate statistical significance, p<0.05.

**Table 3.** Odds ratios for membership to the low and moderate anxiety trajectory for bivariate and fully adjusted models

| Characteristic | Bivariate models | | | Fully adjusted models | | |
|---|---|---|---|---|---|---|
| | OR | 95% CI | *p*-value | OR | 95% CI | *p*-value |
| **Low anxiety** | | | | | | |
| PRH status | | | | | | |
| Unpaid non-helper | *Ref.* | *Ref.* | | *Ref.* | *Ref.* | |
| **Paid PRH** | **3.51** | **1.61, 7.65** | **0.002** | **3.17** | **1.33, 7.52** | **0.009** |
| Paid non-helper | 1.09 | 0.39, 3.05 | 0.9 | 0.81 | 0.25, 2.70 | 0.7 |
| Unpaid PRH | 0.58 | 0.29, 1.15 | 0.12 | 0.86 | 0.40, 1.85 | 0.7 |
| SOC | | | | | | |
| **SOC total** | **1.09** | **1.06, 1.12** | **<0.001** | **1.06** | **1.02, 1.10** | **0.002** |
| Gender | | | | | | |
| Male | | | | *Ref.* | *Ref.* | |
| **Female** | | | | **0.49** | **0.26, 0.92** | **0.027** |
| Non-binary/not disclosed | | | | 0.51 | 0.10, 2.55 | 0.4 |
| Age | | | | 1.02 | 0.98, 1.05 | 0.4 |
| Years in school | | | | 1.01 | 0.96, 1.06 | 0.8 |
| **Number of traumatic events** | | | | **0.81** | **0.71, 0.92** | **0.001** |
| SPS total | | | | 1.03 | 0.99, 1.07 | 0.2 |
| PACT Score | | | | 0.84 | 0.44, 1.60 | 0.6 |
| Years in Greece | | | | 0.95 | 0.90, 1.00 | 0.067 |
| **Moderate anxiety** | | | | | | |
| PRH status | | | | | | |
| Unpaid non-helper | *Ref.* | *Ref.* | | *Ref.* | *Ref.* | |
| Paid PRH | 1.27 | 0.60, 2.68 | 0.5 | 1.32 | 0.60, 2.91 | 0.5 |
| Paid non-helper | 0.85 | 0.36, 2.05 | 0.7 | 0.92 | 0.35, 2.37 | 0.9 |
| Unpaid PRH | 0.82 | 0.49, 1.36 | 0.4 | 0.93 | 0.53, 1.62 | 0.8 |
| SOC | | | | | | |
| **SOC total** | **1.05** | **1.02, 1.07** | **<0.001** | **1.03** | **1.00, 1.06** | **0.027** |
| Gender | | | | | | |
| Male | | | | *Ref.* | *Ref.* | |
| Female | | | | 1.09 | 0.66, 1.81 | 0.7 |
| Non-binary/Not disclosed | | | | 0.49 | 0.13, 1.86 | 0.3 |
| Age | | | | 0.99 | 0.97, 1.02 | 0.6 |
| Years in school | | | | 0.97 | 0.93, 1.01 | 0.15 |
| **Number of traumatic events** | | | | **0.88** | **0.79, 0.97** | **0.008** |
| SPS total | | | | 1.01 | 0.99, 1.04 | 0.3 |
| PACT Score | | | | 1.11 | 0.75, 1.65 | 0.6 |
| Years in Greece | | | | 0.97 | 0.93, 1.02 | 0.2 |

*Note.* CI, confidence interval; GAD-7, generalized anxiety disorder scale; OR, odds ratio; PACT, Perceived Ability to Cope With Trauma; PHQ-9, Patient Health Questionnaire; PRH, Peer Refugee Helper; Ref., reference; SOC-13, sense of coherence; SPS, Social Provisions Scale. Bold values indicate statistical significance, p<0.05.

payment has been associated with higher engagement (Barber et al., 2008). Payment may also correlate with other variables that may be beneficial for the PRHs. For example, organizations that offer paid contracts may have more resources and stability in their work environment. Woodward et al. (2022) found that providing financial incentives could help with the retention of PRHs, thus leading to a more stable work environment. Maybe most significantly, payment maintains livelihood and is especially important for population groups that have few other means for surviving (International Federation of Red Cross and Red Crescent Societies, 2015).

Payment status in PRH work holds a unique role in predicting anxiety and depression trajectories. This may be because payment could be associated with job security that includes stable working hours, clear role boundaries and increased access to essential resources. Poorer working conditions have been found to negatively affect PRH mental health in several studies (Ali et al., 2024; Lavdas et al., 2024b). Sangraula et al. (2024) further suggest that acknowledging the contribution of non-specialist workers with financial compensation may be associated with increased job satisfaction and quality of service provision.

As for SOC, every 1 unit of increase augmented the probability of following a low depression and anxiety trajectory by 5 and 9%, respectively. This finding suggests that the ability to be aware of the

challenges and the resources available, and to mobilize such resources, may help reduce anxiety and depression symptoms. SOC retained its significance when comparing moderate to high anxiety trajectory membership, with a 1 unit increase in SOC, increasing by 3% the probability for moderate, compared to high anxiety trajectory. This is in line with the results presented by Lavdas et al. (2024a) showing that every 1 unit of increase in SOC was associated with a reduction of 0.15 and 0.16 for anxiety and depression scores, respectively. Our findings are further consistent with the work of Veronese and Pepe (2017) who found that SOC protected mental health predominantly for those aid workers without a specialist background, as is the case with PRHs in our study. Task-sharing interventions are associated with concrete roles for non-specialist providers, which include PRHs (Bryant, 2023), and may be associated with competency-based training, which contributes to the development of appropriate "knowledge, attitude and skills" (Pedersen et al., 2023, 2), and supportive supervision that safeguards well-being (Abujaber et al., 2022). As SOC is associated with mental health outcomes in our study and relates to how PRHs might cope with challenges, it is important to consider ways to strengthen SOC.

In our study, we found that higher levels of exposure to potentially traumatic events reduced the probability of following a low anxiety and low depression trajectory. These findings partially align with the cross-sectional study of the T1 data (Lavdas et al., 2024a) where the effect of trauma exposure attenuated in the fully adjusted model. The results of our study reflect that PRHs may have experienced adversity before their flight, as well as during their journey, which may negatively impact PRHs' emotional well-being and worsen mental health (Surkan et al., 2024; Lavdas et al., 2024b). This highlights the importance of providing effective psychosocial interventions that may address mental health symptoms following trauma exposure.

Being a woman has been consistently associated with a higher prevalence of anxiety and depression in the general population (Kuehner, 2017), as well as in conflict-affected populations (Charlson et al., 2019; Bendavid et al., 2021) and within various groups of aid workers (Martinmäki et al., 2023; Sangraula et al., 2024). However, there are unique challenges for refugee women engaged in PRH roles, shaped by a complex interplay of individual and contextual factors through an intersectionality framework. Among PRHs in Greece, women reported adverse reactions from their peers, such as sexual harassment, that impacted negatively on their well-being (Lavdas et al., 2024b). In their accounts, women experienced that it was their responsibility to deal with GBV and reported a lack of institutional support in their coping efforts (Lavdas et al., 2024b). Other women engaged as non-specialist providers in task-sharing interventions further confirm GBV experienced during their work (Sangraula et al., 2024). Other factors that may explain higher burden among women in the humanitarian field could be related to family functions and obligations (Al-Krenawi and Bell, 2022). Women are often primary caregivers and may face increased pressure to maintain family cohesion, care for children and elderly relatives and navigate disrupted family dynamics in displacement settings. These heightened burdens, often without adequate support, can exacerbate stress and emotional distress. Discrimination based on intersecting identities has been associated with increased adverse effects for physical and mental health, and this impact is not simply the sum of separate discrimination forms (Friesinger et al., 2025). Being a woman, experiencing labor precarity and having experienced a wide range of potentially traumatic events, including forced displacement, leads to unique challenges that should be addressed comprehensively at an individual, community and policy level (United Nations Network on Racial Discrimination and Protection of Minorities, 2022).

## Strengths and limitations

Our longitudinal study has several strengths. First, by employing a longitudinal design, our study provides valuable insights into the development of anxiety and depression symptoms over time for PRHs. Second, we recruited a hard-to-reach population working under precarious conditions, allowing us to examine the unique challenges and factors that influence PRH mental health. Third, we applied a data-driven approach, using LGMM to capture variability both within and between groups and to identify latent subgroups with similar mental health trajectories. Fourth, we used well-established instruments to measure our primary variables of interest, such as anxiety, depression and SOC.

Our study has several limitations that should be acknowledged. First, not all participants were included in every time point, which may have impacted the overall robustness of our findings. Second, attrition bias should be acknowledged, and our results must be interpreted with caution. Participants lost to follow-up exhibited higher baseline depression and anxiety scores, and our sample included only those with at least two time points. This may have led to an underrepresentation of individuals with more severe symptoms in the final trajectory estimates, which may also have impacted the trajectory shapes. Third, the use of a convenience sample introduces potential selection bias, limiting the generalizability of our results to other PRHs working in diverse contexts. Fourth, the moderate entropy observed in our models suggests increased variability in participant trajectories, which may affect the interpretability and stability of the identified subgroups. Fifth, our observational design may further limit our potential to draw conclusions regarding the causality of the associations between SOC and mental health symptoms. Although we tested SOC as a predictor of trajectory membership, it is possible that the relationship is bidirectional between SOC and mental health outcomes. For example, negative cognitions in depression and social isolation could lower SOC, while higher SOC may, in turn, protect against poor mental health. Future studies could examine SOC longitudinally, explore its mediating role and compare SOC levels between PRHs and non-helpers. Lastly, we cannot argue whether payment by itself is what makes a difference, or whether it stood in as a proxy for broader structural and organizational factors (e.g., type of contract, level of pay and access to services). Further research should examine payment within this wider context of humanitarian work conditions for PRHs.

## Conclusion

Our findings suggest an association between mental health and PRHs' work environment and have three concrete implications. First, we found that SOC was associated with lower anxiety and depression symptoms over time. SOC is a resource that could be further enhanced through integrating task-sharing interventions in the humanitarian field, including structured roles in MHPSS, competency-based training, as well as supportive supervision. For instance, clearly defined roles and responsibilities may improve individuals' understanding of their tasks, promoting a sense of predictability and control (comprehensibility). Competency-based

training may enhance individuals' perceived ability to manage challenges effectively (manageability), while supportive supervision can provide ongoing guidance and emotional support, reinforcing their sense of purpose and engagement (meaningfulness). Together, these strategies may help build resilience and improve mental well-being among practitioners working in high-stress humanitarian settings. Second, we found that payment among PRHs was associated with the low anxiety trajectory, while being unpaid was associated with the high depression trajectory. Thereby, offering adequate financial incentives and paid contracts could be beneficial for their mental health. Third, our findings related to gender indicate that there is a need to prioritize comprehensive policies and practices that promote mental health among female PRHs, ensuring their safety and well-being in the workplace.

**Open peer review.** To view the open peer review materials for this article, please visit http://doi.org/10.1017/gmh.2025.10068.

**Supplementary material.** The supplementary material for this article can be found at http://doi.org/10.1017/gmh.2025.10068.

**Data availability statement.** Ethics approval was contingent on the storage of research data in the secure facilities at our research institutions. For that reason, we cannot provide the data as Supplementary Material or transfer it to data repositories. Any individual requests for access should be addressed to the corresponding author.

**Acknowledgements.** The authors would like to thank the following organizations that were actively engaged with the research: Médecins Sans Frontières Greece - MSF Experts by Experience, A Drop in the Ocean (DiH), Association for Regional Development and Mental Health (EPAPSY), ECHO100PLUS, Babel Day Centre (mental health unit for migrants and refugees) and the Afghan Migrant and Refugee Community in Greece. The authors would like to further thank their reference group, consisting of stakeholders in the humanitarian field in Greece. The authors express their gratitude for the collaboration with Mahdia Hossaini, Farahnaz Faiz, Parsa Saidi, Nader Turkmani and Jan Ali, who contributed with their language and culture expertise in formulating the survey and facilitating data collection in their study. Finally, the authors would like to thank Berge Osnes and Diana Czepiel for the meaningful exchanges on the analysis, along with Mathea Homme and Martika I. Brook for providing reflective input.

**Author contribution.** ML: Conceptualization, data curation, funding acquisition, investigation, visualization, formal analysis, project administration, writing – review and editing, writing – original draft. GMS: Conceptualization, project administration, methodology, supervision, writing – review and editing. MS: Supervision, methodology, visualization, formal analysis, writing – review and editing. TH: Supervision, methodology, visualization, formal analysis, writing – review and editing. TB: Conceptualization, project administration, methodology, visualization, formal analysis, supervision, writing – review and editing.

**Financial support.** The main funding was provided through the PhD project that the first author conducted at the University of Bergen, Department of Psychosocial Science. Supplementary funds were further provided by the Meltzer Research Fund, Bergen, Norway (Application numbers 27563 and 28343).

**Competing interests.** The authors declare none.

**Ethics statement.** The Norwegian Regional Committee for Medical and Health Research Ethics Vest (REK, number 422160) approved the study. The study did not meet the requirements for review by the Greek National Ethics Committee, as it only considers studies involving pharmaceutical drugs or medical devices.

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
