## [Reviewer Report]

Overall a really interesting and valuable paper. I have a few suggestions:

1. Clarity of language: high/low depression/anxiety trajectory sounds like more of a research term than a clinical one, and may not be clear to readers what it’s referring to. E.g in lines 29 and 30, an explanation could be added, such as “i.e. to score more highly on a measure of depression”. As a clinician reading this at the beginning of the paper, I wasn’t clear if high depression trajectory meant high likelihood of developing depression, or high likelihood of worsening symptoms over time.

2. The aims of the study don’t mention traumatic events and their relation to the trajectories - this is an interesting element that it should be set out at the beginning of the paper as well as in the results/discussion.

3. In the introduction (line 79) you mention that being a peer refugee helper is protective to mental health (as well as having some adverse effects), and imply that that mechanism may be via improving Sense of Coherence, yet I note that in your results you haven’t compared SOC scores between helpers and non-helpers. Readers would be interested to know whether this is the case.

4. It would be important to acknowledge in the paper that there is likely a bidirectional relationship between SOC and particularly depression, although likely also anxiety. Both the high depression and high anxiety trajectory groups commence the study with above diagnostic threshold symptoms, making it impossible to know whether the SOC preceded or followed the depression/anxiety. I think it’s quite likely that a depressive illness could lead to a reduction in SOC, given the negative cognitions etc.

5. Clarity of language in lines 460 and 461: “Second, we found that payment among PRHs contributes to their anxiety and depression scores” - this sounds to me as if the payment is making the anxiety/depression worse rather than the other way around!

---

## [Reviewer Report]

1. Summary and Overall Impression

This manuscript offers a well-executed and timely contribution to global mental health and humanitarian research. Through longitudinal analysis of depression and anxiety trajectories among Peer Refugee Helpers (PRHs) and non-helpers in Greece, the authors generate valuable evidence around the impacts of payment status, sense of coherence (SOC), gender, trauma exposure, and social support on mental health outcomes.

The study’s methodological rigour, particularly its use of Latent Growth Mixture Modelling (LGMM), is commendable. The authors are careful and reflexive in describing their tools, limitations, and context. Of particular strength is the meaningful engagement of multilingual research assistants with lived experience, as well as the thoughtful inclusion of a stakeholder reference group. These elements enhance the paper’s ethical and applied relevance.

The paper’s framing, while already compelling, could be further strengthened by deepening the conceptual treatment of labour precarity, expanding the discussion of intersectional gendered risk, and providing additional detail on ethical and trauma-informed research processes. These suggestions are offered with the aim of supporting both this study and the broader field.

2. Prioritised Major Comments

Essential Revisions (to address in this paper)

Clarify the relationship between payment and mental health outcomes

The finding that unpaid PRHs are more likely to follow high depression trajectories is central to the paper. However, it remains unclear whether this is directly attributable to payment or whether payment serves as a proxy for broader dimensions of protection, formality, and organisational support.

Suggestion: Clarify how “unpaid” was operationalised (e.g. were these formal volunteer roles, informal community roles, roles without defined support or expectations?). Consider briefly situating payment within a wider context of humanitarian labour conditions (e.g. contract type, supervision, access to services).

Strengthen the framing of gender as a structural and intersecting factor

The analysis shows that women were less likely to follow low-symptom trajectories, but the discussion frames gender largely as an individual demographic risk factor. This misses the opportunity to situate these findings in relation to structural gender inequalities—including exposure to sexual harassment, caregiving pressures, or constrained mobility in the humanitarian workplace.

Suggestion: Reframe gender as a system of power, shaped by intersectional factors such as displacement status, age, and caregiving roles. Consider referencing relevant humanitarian gender analyses or the concept of intersectionality (Crenshaw, 1991) to anchor this shift.

Acknowledge and briefly interpret loss to follow-up

The authors note that participants lost to follow-up had higher depression scores at baseline, which raises questions about potential underestimation of symptom severity or risk in the final trajectories.

Suggestion: Expand slightly on how this differential attrition may have affected results, and whether any sensitivity analyses or interpretive caveats were applied.

Recommended Enhancements (optional additions)

Reposition SOC as relational and modifiable

The paper treats SOC primarily as an internal resource, but findings suggest that SOC may be shaped by supervision, training, safety, and peer support.

Suggestion: Consider briefly repositioning SOC as a context-sensitive and modifiable construct, which can be strengthened through program design and organisational support.

Provide clearer programmatic implications

The conclusion is strong, and also could go further in identifying what a “minimally ethical” PRH engagement might look like in practice. This is a strength of the paper that could be made more actionable.

Suggestion: A short paragraph summarising practical implications, such as the importance of role clarity, paid contracts, mental health support, and safeguarding, would enhance the paper’s utility to implementers.

Comment on the translation process and ethical inclusion of research assistants

The involvement of research assistants with lived experience is a methodological strength. However, it is unclear how their wellbeing was supported, and how the translation process addressed potential cultural stigma, affirming mental health language, or other identity-based risks.

Suggestion: If feasible, briefly note whether trauma-informed supports (e.g. debriefing, flexibility) were in place for research assistants, and whether mental health-related terminology was discussed and adapted for affirming language. This is not to suggest a flaw, but rather to encourage good practice documentation that others can learn from.

Note: If considerations such as multi-identity stigma were outside the scope of this study, it is perfectly valid to simply note that future work might explore these dimensions more deeply.

3. Minor and Stylistic Suggestions

Abstract clarity: Consider rephrasing “unpaid PRHs were associated with increased probability of following a high depression trajectory (OR = 0.55)” as this implies lower odds. A clearer phrasing might be: “Unpaid PRHs were significantly less likely to follow a low depression trajectory.”

Terminology: Consider balancing the use of legal classifications (“refugees and asylum seekers”) with more inclusive phrasing such as “people with lived experience of displacement.”

Figures: Ensure that trajectory plots clearly label high/moderate/low categories and reference clinical thresholds where relevant.

Language tightening: Some sections (particularly citations to Lavdas et al. 2024) could be trimmed slightly for clarity and flow.

Ethical procedures: It would be helpful to briefly note whether any referral pathways were in place for participants who disclosed distress.

4. Final Recommendation

This is a thoughtful and impactful study, grounded in methodological rigour and ethical intent. The suggestions offered here are intended to support the authors in strengthening the framing, expanding ethical transparency, and improving clarity around working conditions and programmatic implications. These are relatively minor revisions that, once addressed, will significantly enhance the paper’s contribution to research and practice.

---

## [Editor Report]

Thank you for submitting this well-written and timely manuscript. Reviewers commended the study’s methodological rigor, ethical approach, and meaningful engagement of research assistants with lived experience. As you will see in their detailed comments, both reviewers have provided some suggestions for improving the manuscript including strengthening the framing around gender, addressing loss to follow-up and clarifying some of the conceptual elements of the paper.

---

## [Reviewer Report]

Many thanks for outlining your changes so clearly. I’m satisfied that all of my comments have been resolved. I have just a couple of minor points relating to the new additions:

Lines 105-108:

I suggest using the words “personal experiences of traumatic events” rather than “own”, it just reads a little better.

Lines 451-456:

I would add that PRHs may have experienced adversity prior to as well as during their journeys, as in my experience many people choose to leave their countries because of potentially traumatic events experienced there.

Line 455:

I would use the phrase “worsen mental health” rather than “increase poor mental health” as it is clearer on reading.

Line 458:

I would suggest simply deleting “including anxiety and depression symptoms” because quite often in PTSD, treating the PTSD directly leads to resolution of anxiety and depressive symptoms, and therefore those symptoms wouldn’t necessarily be directly addressed in practice (even though they would improve).

---

## [Editor Report]

Thank you for your thorough revision of the manuscript. The reviewer noted that you have adequately addressed all of their major comments from the previous round, and I agree that these changes have strengthened the paper. At this stage, there are a few minor edits for your consideration. We kindly ask that you review and incorporate these minor changes as appropriate before we proceed further.

---

## [Editor Report]

All comments have been thoroughly addressed. Thank you for your thoughtful and careful revisions, and for submitting your work to Global Mental Health.